



# 1 Observations of the atmospheric boundary layer structure

# 2 over Beijing urban area during air pollution episodes

Linlin Wang[1,2], Junkai Liu[1,5], Zhiqiu Gao[1*], Yubin Li[2], Meng Huang[2], Sihui Fan[2],
Xiaoye Zhang[3], Yuanjian Yang[2], Shiguang Miao[4], Han Zou[1], Yele Sun[1], Yong
Chen[1], Ting Yang[1]
[1]State Key Laboratory of Atmospheric Boundary Layer Physics and Atmospheric Chemistry
(LAPC), Institute of Atmospheric Physics, Chinese Academy of Sciences, Beijing 100029, China
[2] Collaborative Innovation Centre on Forecast and Evaluation of Meteorological Disasters, School
of Atmospheric Physics, Nanjing University of Information Science and Technology, Nanjing,
210044, China
[3] Chinese Academy of Meteorological Sciences, Beijing, 100081, China
[4] Institute of Urban Meteorology, China Meteorological Administration, Beijing, 100081, China
[5]University of Chinese Academy of Sciences, Beijing 100049, China

## 15 **Abstract**

We investigated the interactions between the air pollutants and the structure of
urban boundary layer (UBL) over Beijing by using the data mainly obtained from the
325-m meteorological tower and a Doppler wind lidar during 1–4 December, 2016.
Results showed that the pollution episodes in this period could be characterized by
low surface pressure, high relative humidity, weak wind, and temperature inversion.
Compared with a clean daytime episode that took place on 1 December, results also
showed that the attenuation ratio of downward shortwave radiation was about 4%, 23%
and 78% at 1200 local standard time (LST) on 2–4 December respectively, while for
the net radiation ($R_n$) attenuation ratio at the 140-m level of the 325-m tower was 2%,
24%, and 86%. The large reduction in $R_n$ on 4 December was not only the result of the
aerosols, but also clouds. Based on analysis of the surface energy balance at the

*Corresponding author: Dr. Zhiqiu Gao, zgao@mail.iap.ac.cn



140-m level, we found that the sensible heat flux was remarkably diminished during
daytimes on polluted days, and even negative after sunrise (about 0720 LST) till 1400
LST on 4 December. We also found that heat storage in the urban surface layer played
an important role in the exchange of the sensible heat flux. Owing to the
advantages of the wind lidar having superior spatial and temporal resolution, the
vertical velocity variance could capture the evolution of the UBL well. It clearly
showed that weak vertical mixing caused the concentrating of pollutants, and that
vertical mixing would also be weakened by a certain quantity of pollutants, and then
in turn worsened the pollution further. Compared to the clean daytime on 1 December,
the maximums of the boundary layer height (BLH) reduced about 44% and 56% on
2−3 December, when the average $PM_{2.5}$ ($PM_1$) concentrations in afternoon hours
(from 1200 to 1400 LST) were 44 (48) µg m$^{-3}$ and 150 (120) µg m$^{-3}$. Part of these
reductions of the BLH was also contributed by the effect of the heat storage in the
urban canopy.

# 1  Introduction

In recent years, fine particulate matter (PM) pollution events in the atmospheric
boundary layer (ABL), i.e., involving particles with diameters $\leq 2.5$ µm ($PM_{2.5}$), have
occurred frequently in urban areas, thus emerging as a serious environmental issue in
China. The Beijing-Tianjin-Hebei (BTH) metroplex region is one of the most
seriously affected areas in China with respect to air pollution. The main hazards or
negative effects of air pollution generally fall into two categories: human health and
transportation. Thus, it is an issue that has attracted considerable public attention and,
accordingly, numerous studies have focused on investigating the sources and
formation mechanisms of air pollution in the BTH region, through numerical
simulation and field observational methods (e.g., Wang et al., 2013; Sun et al., 2014;
Ye et al., 2016; Li et al., 2017; Han et al., 2018).
Beijing, the main city of the BTH region, has experienced several high-impact,
persistent, and severe air pollution episodes in recent years, with notable examples


having taken place in January 2013, October and November 2014; December 2015
and 2016, and January 2017. Beijing is located on the North China Plain (NCP), and
is surrounded by the Yan and Taihang Mountains from north to west. Therefore,
Beijing is frequently affected by thermally induced mountain-plain wind circulation
over the NCP, which contributes to the transportation of air pollution in Beijing (Liu
et al., 2009; Hu et al., 2014; Chen et al., 2017; Zheng et al., 2018). In addition, it is
well recognized that high levels of anthropogenic emissions and rapid formation of
secondary aerosol are key factors leading to the frequent occurrence of severe haze
episodes (Li et al., 2017). More importantly, these interactions on local and large
scales are associated with the meteorological conditions (Sun et al., 2013; Yang et al.,
2018). Previous studies have reported that heavy pollution in Beijing is highly related
to unfavorable local weather conditions, such as weak wind, strong temperature
inversion, high relative humidity (RH) and low surface pressures (Zhang et al., 2014,
Liu et al., 2017, Li et al., 2018).

Many studies have also suggested that the structure of the urban boundary layer

(UBL), in particular wind, turbulence and stability, had strong influences on the
occurrence, maintenance, vertical diffusivity of air pollutants (Han et al., 2009; Zhao
et al., 2013). For instance, emissions of air pollution in urban areas lead to a buildup
of pollutant concentrations due to reduced mixing and dispersion in UBL (Holmes et
al., 2015). An analysis of the dramatic development of a severe air pollution event on
November 2014 in the Beijing area revealed that turbulent mixing played an
important role in transporting the heavily polluted air and $PM_{2.5}$ oscillations (Li et al.,
2018). The vertical profiles of wind and temperature along with the BLH are the main
factors affecting turbulence diffusion. Moreover, the BLH is also a key variable in
describing the structure of UBL, and in predicting air-pollution (Stull, 1988; Miao et
al., 2011; Barlage et al., 2016). Miao et al. (2018) found that the concentration of
$PM_{2.5}$ anti-correlates with the BLH. In addition, air pollutants also can modulate
radiative transfer processes through the scattering, reflection and absorption of
shortwave radiation and the reflection, absorption and emission of longwave radiation
(Dickerson et al., 1997; Stone et al., 2008; Wang et al., 2014). In response to reduced



solar radiation, the cooling of surface air temperature can lead to strong temperature
inversion in the near-surface layer, which can increase the atmospheric stability and
prolong the accumulation of pollution because of the existence of this stable boundary
layer (Barbaro et al., 2013; Che et al., 2014; Gao et al., 2015). A positive feedback
loop in which more aerosol loading leads to a more stable atmospheric boundary layer
(ABL), enhanced accumulation of pollutants within the ABL, and a more polluted and
hazier atmosphere, was described by Zhang et al. (2013; 2018). It is also found that
the further worsened meteorological conditions caused by cumulated aerosol pollution
dormant subsequently occurred "explosive growth" of $PM_{2.5}$ mass that often appears
in the late stage of heavy aerosol pollution episode in Beijing-Tianjin-Hebei area in
China (Zhong et al., 2017).

Although many studies have provided various interesting findings, consensus has

not been reached on the pollutant transport mechanism and the nature of the
interactions between the air pollution and the structure of the UBL, mainly due to a
lack of reliable and detailed field measurements and the complex properties of the
UBL. Additionally, as mentioned above, there are several factors that affect the
occurrence of urban air pollution, which can lead to different pollutant transporting
mechanism characteristics for different pollution events. Therefore, taking a severe
heavy pollution event occurred during 1–4 December, 2016 in Beijing as an example,
we will aim to investigate evolution characteristics of ABL structure and further
explore the interaction between the structure of the UBL and the air pollution by using
the field data collected from a 325-m meteorology tower in Beijing urban area, as
well as from a Doppler wind lidar and a dual-wavelength (1064 and 532 nm)
depolarization lidar. During this pollution episode, the $PM_{2.5}$ concentration rapidly
increased from about 100 μg m$^{-3}$ to approximately 500 μg m$^{-3}$ at 1200 LST on 4
December, which can be considered as a typical case to achieve a better
understanding the formation, transportation, and dispersion mechanisms of the alike
pollution event, as well as the interactions between the air pollution and the structure
of the UBL.

The paper is organized as follows: Section 2 describes the field site, data, and





methods. The overall characteristics of the synoptic pattern and the meteorological
factors related to the development of the pollution event are investigated in Section 3.
The impacts of the evolution of the vertical UBL structure on this pollution episode,
and vice versa—especially the turbulence due to the radiative forcing of aerosols—are
also explored in Section 3. Lastly, the results of the study are summarized in Section

4.

## 2  Materials

### 2.1  Site and data

The main data used in this study were from a tall tower in Bejing, officially known
as "the Beijing 325-m meteorological tower" which is located at an urban site in the
city (39.97°N, 116.37°E; the Beijing "inner-city" site). Within a radius of 5 km of the
tower, buildings of different heights are distributed irregularly in all directions, and
the area is surrounded by four-story to twenty-story buildings with heights of 10 – 60
m (Liu et al., 2017). The surrounding buildings can be seen in Fig.1a. This tall tower
conducts turbulent flux measurements using sonic anemometers (Model Windmaster
Pro, Gill, UK) at three different levels (i.e., 47-m, 140-m and 280-m).
Note that CSAT3 three-dimensional sonic anemometers designed by Campbell
Scientific Inc (USA) at these three levels have been replaced by the Model
Windmaster Pro since 2015, so the turbulence measurements before 2015 used in
previous papers were collected using the CSAT3 sonic anemometers. The new sonic
anemometer experimental setup has been reported by Cheng et al. (2018). Downward-
pointing and upward-pointing pyrgeometers and pyranometers (CNR1, Kipp &Zonen)
are maintained at the same heights as the sonic anemometers to measure
four-component radiation (i.e., incoming shortwave and longwave radiation, and
outgoing shortwave and longwave radiation). Meteorological elements, including
wind speed, wind direction (010C cup anemometers and 020C wind vanes, Metone,
USA), RH and temperature (HC2-S3, Rotronic, Switzerland) are measured at 15
levels (i.e., 8-m, 15-m, 32-m, 47-m, 65-m, 80-m, 100-m, 120-m, 140-m, 160-m,



180-m, 200-m, 240-m, 280-m and 320-m) above ground level. An Aerodyne aerosol
chemical speciation monitor and a high-resolution time-of-flight aerosol mass
spectrometer were deployed at 260-m and ground level, repetitively to measure $PM_1$
mass concentrations at 5-min intervals (Sun et al., 2016).

In addition, wind speed (05103-L, R. M.Young) and temperature (HMP45C,

Vaisala) at the 2.2-m level are measured at a surface station about 20 m south of the
tower. We also used wind data collected above 100 m by a Doppler wind lidar
(Windcube200, Leosphere, Orsay, France) situated on the rooftop of a 8 m high
building. Furthermore, a dual-wavelength (1064, and 532 nm) depolarization lidar
developed by the National Institute for Environmental Studies, Japan, sits on the
rooftop of a 28 m high building (Yang et al., 2017), which provided us with
information 0n aerosols at higher layer. The mass concentrations of $PM_{2.5}$ measured at
the Beijing Olympic Sports Center of the National Air Quality Monitoring Network of
China using Tapered Element Oscillating Microbalance analyzers with hourly
monitored readings, were obtained from the website of China National Environmental
Monitoring Center (http://113.108.142.147:20035/emcpublish).

The three-dimensional sonic anemometers original records (10 Hz) were

processed, prior to analysis using the methods of double rotation (i.e., yaw and pitch
rotations) and linear detrending. Wang et al. (2014) tested a few averaging periods and
found that a 1-h averaging period is reasonble at this urban site. The processing of
turbulence data in our study followed the method described by Wang et al. (2014).

The criterion of threshold carrier-to-noise ratio (CNR) was used to reduce the

effects of invalid data on profiles derived from the Doppler velocities. The data
control process was described in detail by Huang et al. (2017). We calculated the
vertical velocity variance and stream wise wind speed and wind direction over a
30-minute segment.

The dual-wavelength depolarization lidar was used to retrieve the aerosol vertical

structure at a spatially resolved resolution of 6 m and temporally resolved resolution
of 10 s, but only for altitudes in excess of 100 m because of an incomplete overlap
between the telescopic field of view and the laser beam. For this study the raw



temporal resolution of the retrieved aerosol profiles was set at 30-minute. More
details on the lidar instruments and various data processing techniques were provided
by (Yang et al., 2017).
The NCEP FNL (Final) Operational Global Analysis data collected every six
hours, at 0200, 0800, 1400 and 2000 LST, on $1° \times 1°$ grids was used to analyze the
synoptic-scale weather conditions.

## 2.2 Methods

### 2.2.1 Turbulent flux and radiation calculation

The sensible heat and latent heat fluxes were calculated using the
eddy-covariance method:

$$H = \rho C_p \overline{w'T'} \quad (1)$$

$$LE = L_v \overline{w'q'} \quad (2)$$

where $u$, $v$, and $w$ are the streamwise, cross-stream, and vertical velocities (m s$^{-1}$),
respectively from the sonic anemometers; $T$ is the air temperature (K), $\rho_v$ is the water
vapour density (kg m$^{-3}$) and $\rho$ is the air density (kg m$^{-3}$). $C_P$ is the specific heat
capacity at constant pressure (J kg$^{-1}$ K$^{-1}$) .
The one-dimensional SEB is usually formulated as

$$R_n + Q_F = H + LE + G \quad (3)$$

where $H$ is the sensible heat flux from the surface to the adjacent air, $LE$ is the latent
heat flux into the atmosphere associated with evapotranspiration, and $G$ is the ground
and urban canopy heat storage. $R_n$ is the net radiation, which can be described as

$$R_n = DSR - USR + DLR - ULR \quad (4)$$

$DSR$ stands for downward shortwave radiation, $USR$ for upwelling short-wave
radiation, DLR for downward incoming long-wave radiation, and $ULR$ for



upwelling long-wave radiation. $Q_F$ is the anthropogenic heat flux which was omitted
here, because of the absence of accurate energy consumption and traffic flow data.
Therefore the heat storage was calculated as $R_n - H - LE$.

## 2.2.2 Determination of UBL depths


Lidar techniques have become the most valuable and popular systems to detect
the atmosphere because of their higher spatiotemporal resolution. As a result, many
techniques have been developed to determine the BLH by using the remote sensing
instruments, such as radar wind profilers, aerosol lidars, and ground-based microwave
radiometers (Flamant et al., 1997; Emeis et al., 2004, Haman et al., 2012). Remote
sensing is particularly useful in analyzing vertical profiles of turbulence mixing in
UBL, and is generally easier to deploy than radiosondes (Georgoulias et al., 2009).
Recently, the turbulence method to define the BLH has been proposed. The
height of the layer in which vertical velocity variance $\sigma_w^2$ from the Doppler lidar
exceeds a given threshold is considered as the BLH. Previous investigators have given
different values of $\sigma_w^2$ for different underlying surfaces (Tucker et al., 2009, Pearson
et al. 2010). Barlow et al. (2011) defined the mixing height as the height over London,
UK up to which $\sigma_w^2 > 0.1 \text{ m}^2 \text{s}^{-2}$. Here, we select this method of Barlow et al. (2011),
because of the similar urban fraction between central Beijing and London.
The 30-min vertical velocity standard deviation between lidar is
$$\sigma_w = \sqrt{\frac{1}{N-1}\sum_{i=1}^{N}(w_i - \overline{w})} \quad (5)$$

Where N is the record number every 30 minutes, $w_i$ denotes the $i$th vertical velocity
(m s$^{-1}$), and $\overline{w}$ is the mean vertical wind speed.

## 3 Results and discussion


### 3.1 Air pollution episodes in Beijing


As shown in Fig. 1c, the visibility around the 325-m tower at about 1400 LST on
3 December was much lower than that on 1 December. In fact, the visibility decreased




rapidly from 1200 to 1600 LST before sunset (1650 LST) on 3 December,
accompanied by the increasing $PM_{2.5}$ concentration (from 100 µg m$^{-3}$ to 200 µg m$^{-3}$)
at the Olympic Sports Center station and $PM_1$ concentration (from 100 µg m$^{-3}$ to 190
µg m$^{-3}$) at the 325-m tower station (Fig. 2). After sunset, the $PM_{2.5}$ hourly maximum
concentration reached 530 µg m$^{-3}$ at 0200 LST 4 December. The cumulative
explosive growth process of the pollution, starting at 1200 3 December and lasting till
0200 LST 4 December, is defined as cumulative stage (CS).

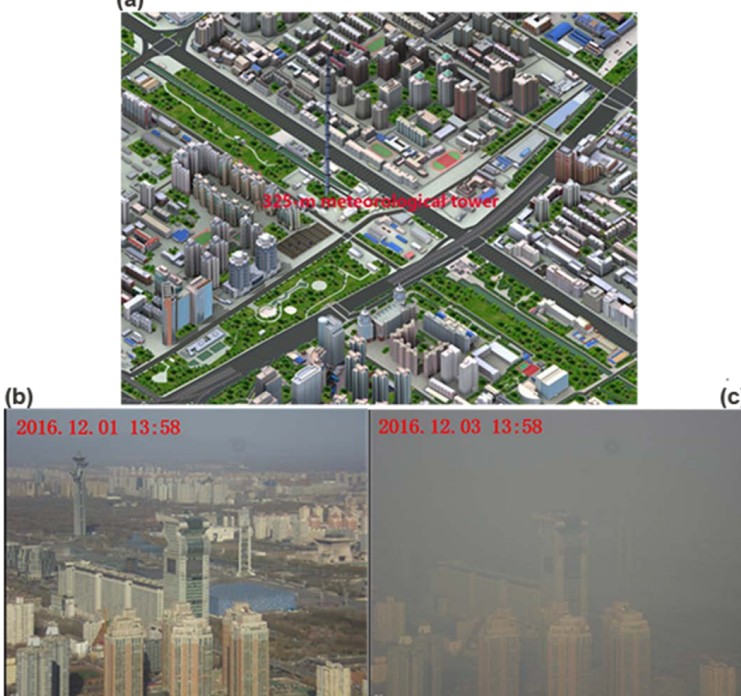


**Figure 1: (a) Three-dimensional graph of the underlying surface around the 325-m tower in**
**Beijing. Photographs of the buildings looking north from the 280-m level of the 325-m tower**
**at 1358 LST (b) 1 December and (c) 3 December, 2016.**



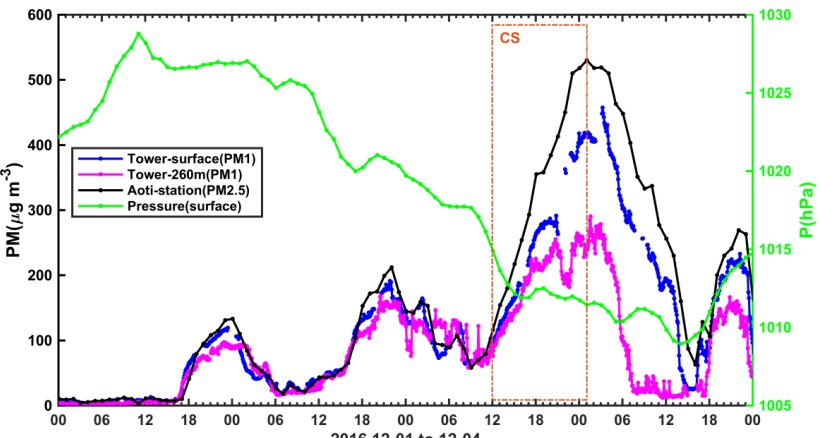


**Figure 2: Temporal variation of the PM$_1$ observed at the surface and the 260-m level of the 325-m tower, PM$_{2.5}$ at Aoti surface station, and surface pressure at the surface station of the IAP, during 1–4 December 2016. (red box: CS )**


The surface pressure measured at the Institute of Atmospheric Physics (IAP)
surface station (Fig. 2) indicated the air quality was getting worse with decreasing
surface pressure. In order to analyze the synoptic background fields for the CS, the
sea level pressure and surface wind field on 3 December are shown in Fig. 3. At 0800
LST, the Beijing region was governed by a saddle type pressure field characterized by
uniform pressure, very weak wind speed and changeable wind direction. The surface
high pressure system over the Bohai and Yellow seas was conducive to the
maintenance of these stagnant meteorological conditions till 1400 LST, which
provided the unfavorable meteorological conditions for the diffusion of air pollutants
and partly led to the CS.

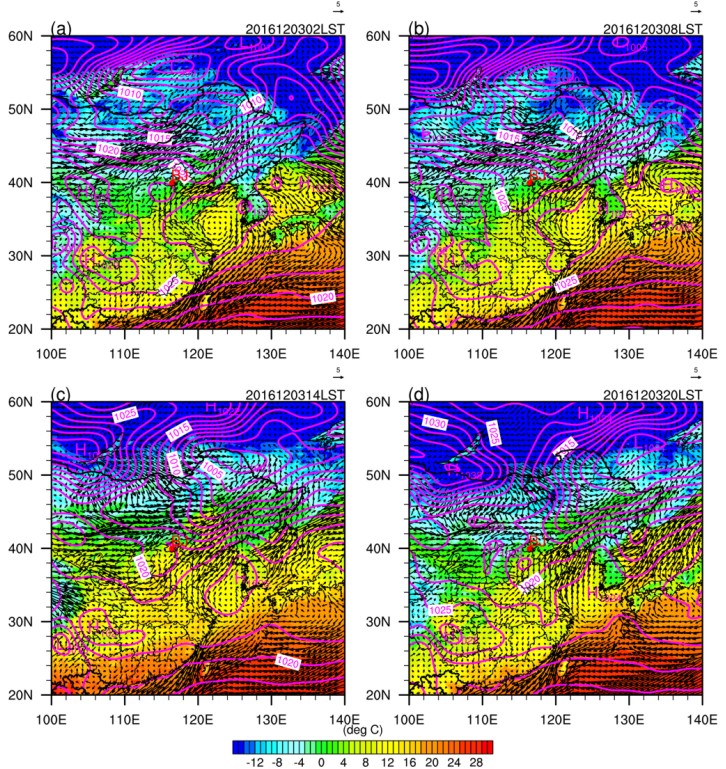

**Figure 3: Distribution of surface pressure and temperature at (a) 0200 LST, (b) 0800 LST, (c) 1400 LST, and (d) 2000 LST 3 December 2016, where the green star marks the location of Beijing (BJ).**

Note that, 3 and 4 December were weekend days without vehicle restrictions, meaning the relatively larger quantities of automobile exhaust were in part responsible for this heavy pollution process.

## 3.2 Meteorological parameters

As shown in Fig.4a, RH was mostly larger than 40% during pollution episodes, and increasing along with the concentrated $PM_{2.5}$ ($PM_1$). Especially during the CS, RH could reach near to 100% at nighttime, which firstly appeared at the levels of 160–220 m and then extended to the lower levels. Meanwhile, the deeper RH (>80%) with higher PM concentrations during the CS was possibly caused by secondary aerosol formation. Due to aerosol cooling force, the $\theta$ at the daytime on 3 December



was much lower than on other days. Temperature inversions were found at all three
nights. Clearly, the wind flow played an important role in the air pollution process.
The Southwesterly wind transported air pollutants from Hebei Province to Beijing on
the first two pollution nights (Fig. 4c). In order to investigate the characteristics of the
UBL structure, the vertical gradients of potential temperature $(\Delta\theta = \theta_2 - \theta_1)$ and
gradient absolute values in wind speed $(|\Delta U| = |U_2 - U_1|)$ were calculated by
using the adjacent two levels as the thermal and dynamic factors (Fig. 5). It was found
that the vertical gradients of wind speed and potential temperature were small because
of strong vertical thermal mixing during daytime, whereas they were large at
nighttime due to weak vertical mixing. The values of $\Delta\theta$ and the duration of $\Delta\theta > 0$
increased day by day, meaning the thermal stability strengthened with worsening
polluted days. A long-term existence of temperature inversion near the surface could
be found till 1200 LST 4 December, associated with extremely steady stability. This
stable surface stratification resulted in the suppressed diffusion of air pollutants at the
surface, causing a dissipation lag for $PM_1$ at the surface compared to the case at the
260-m level (shown in Fig. 2).

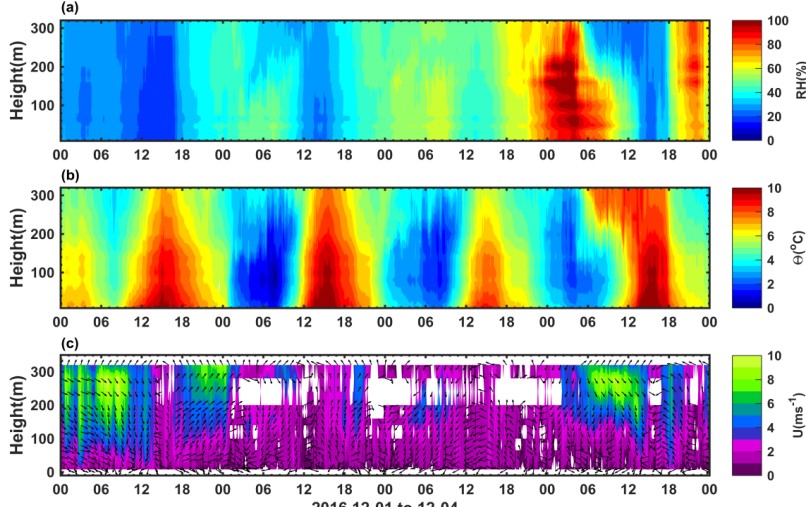


**Figure 4: Vertical evolution of (a) relative humidity, (b) virtual temperature, and (c) wind**
**speed and wind vectors (arrows), observed at 15 levels of the 325-m tower during 1–4**
**December 2016.**





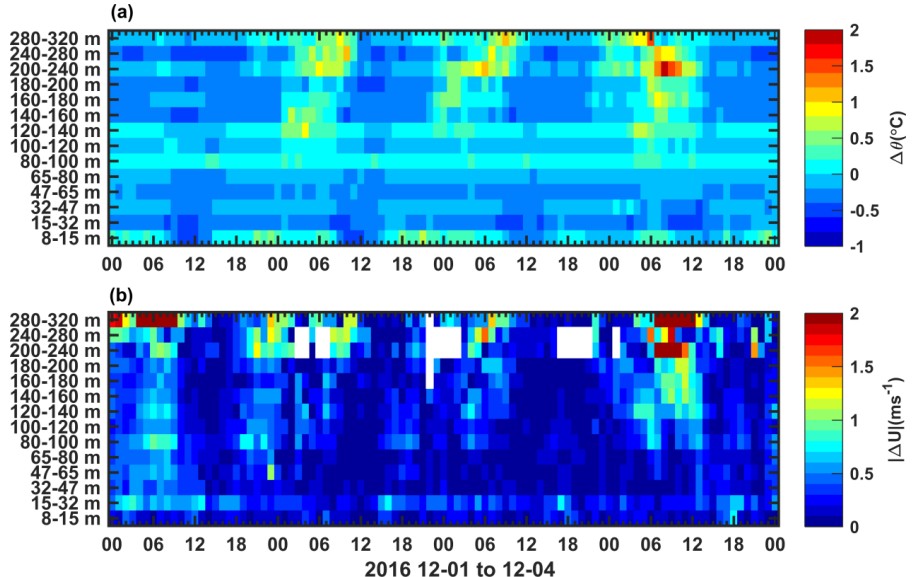

**Figure 5: Vertical evolution of (a) vertical gradients of relative potential temperature and (b)**
**vertical gradients of zonal wind speed, based on observations at 15 levels of the 325-m tower**
**during 1–4 December 2016.**
Owning to the limited height of the tower, the wind profile above several
hundred meters collected by the Dopper lidar (Fig. 6) can be used to further
investigate the association between the wind flow and air pollution process. On 1
December, the air quality was good before noon and there was strong northwest wind
(mostly around 10 m s$^{-1}$) at 200–1000 m levels above the ground (ATG). In our case,
notably, a low-level jet (LLJ) established after sunset, with the jet core at 300–500 m
ATG, and the maximum wind speed was around 10 m s$^{-1}$ at about 2400 LST. We can
see the PM$_{2.5}$ /PM$_1$ concentration was starting to increase after sunset with the
maximum PM$_{2.5}$ concentration (120 μg m$^{-3}$) observed at 2400 LST, and then
decreased with the gradually weakened LLJ, which suggests this southwesterly LLJ
transferred polluted air from the south by advection to Beijing before midnight. A
previous study also reported that presence of an LLJ can increase the surface pollution
through horizontal advection (Hu et al., 2013). Besides the horizontal advection, LLJ
also can generate vertical mixing due to the wind shear with large $|\Delta U|$ ($> 1$ ms$^{-1}$).
Once the northern maintain flow generated, the LLJ became weaker ($< 5$ ms$^{-1}$) in the





early morning on 2 December, and then the vertical mixing generated by the
weakened LLJ changed to the dominated term which made an important contribution
to the mixing of the pollutants at the dissipated period. Chen et al. (2018) also pointed
out that a northerly weak LLJ noticeably reduced the PM concentration in urban
Beijing. As a result, the presence of an LLJ has an indispensable effect on the process
of the air pollution in the nocturnal boundary layer (NBL).

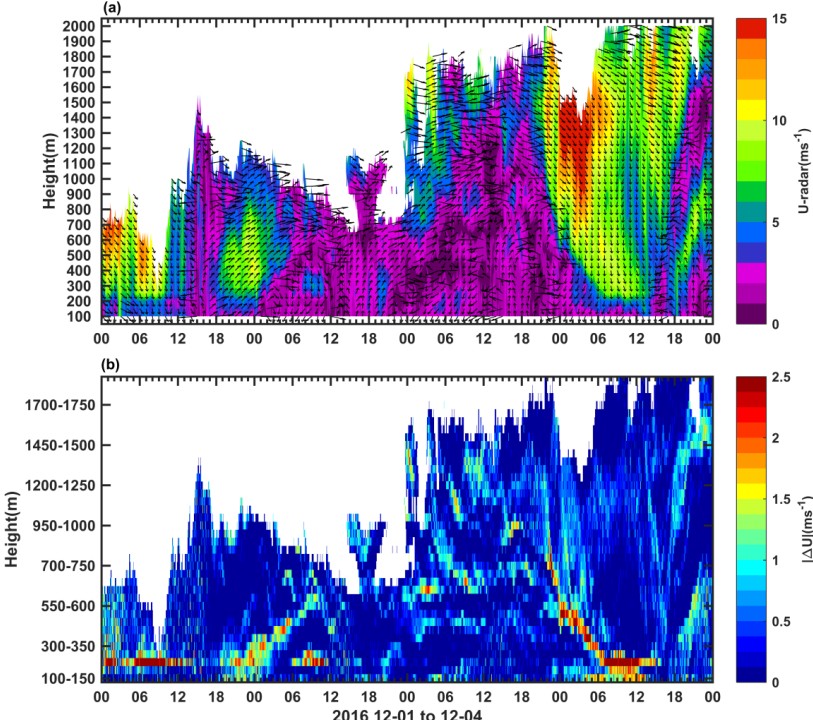


**Figure 6: Vertical evolution of (a) wind speed and (b) vertical gradients of wind speed, based**
**on Doppler wind lidar observations during 1–4 December 2016.**
We can also see that the $PM_1$ concentration at the 260-m level started to
decrease at 0200 LST 2 December which was about two hours later than $PM_1$ at the
ground level. This could be explained that the gradually deep and clean northwest
mountain-plain wind occurred first below 100 m ATG, and then reached the upper
level. On 2 December, the wind below 1 km was dominated by speeds of around 2 m
s$^{-1}$ from 0600 to 2200 LST. The weak northerly winds did not fully disperse the air
pollutants before the noon. Meanwhile, after the transition time on 1300 LST,



southerly winds existed and brought polluted air from the south, and then the air
quality became worsened, and the maximum $PM_{2.5}$ concentration (210 μg m$^{-3}$)
occurred at 2200 LST. Compared to early morning on 2 December, the wind below
600 m was weaker and the vertical gradients (Fig. 6b) were much smaller, meaning
mechanical turbulence (vertical mixing) was extremely weak. Thus, there is no
dramatic reduction in the air pollution before sunrise on 3 December, and then the CS
began at noon when the wind speeds were mostly lower than 3 m s$^{-1}$ below 1 km ATG,
because of the saddle–type pressure-field background (Fig. 3).

## 3.3   SEB characteristics

Radiation from the sun is the most important driver of the development of the
UBL. Various climatic changes within urban ABL are driven by the SEB, which
distributes the energy by radiation, convection and conduction between a facet (Oke et
al., 2017). Therefore, the SEB, described as Eq. 3, is a fundamental aspect
contributing to our understanding of the variations in the UBL.





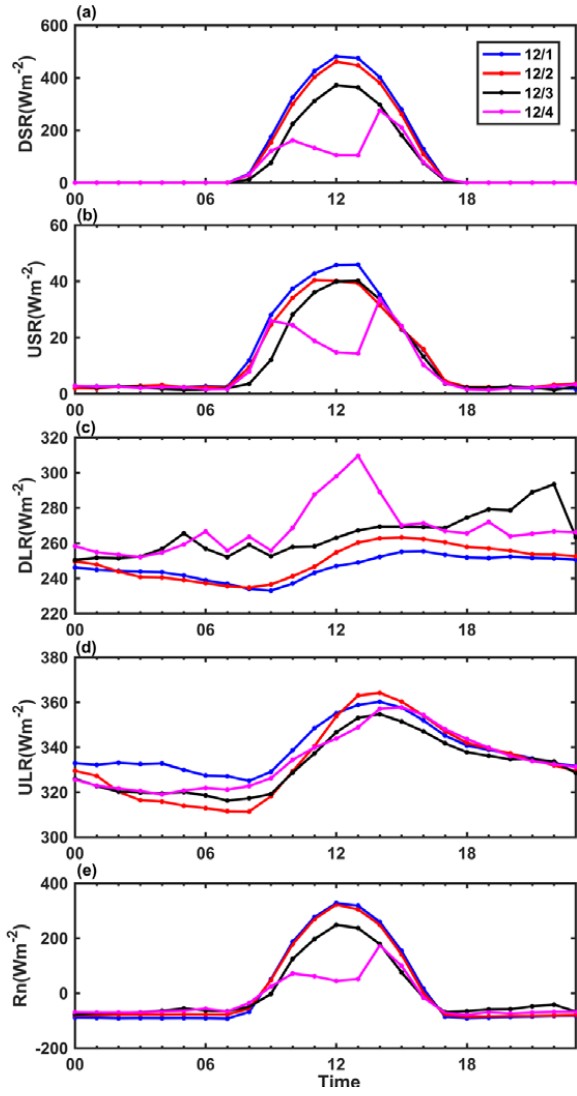


**Figure 7: Diurnal cycle of (a) downward shortwave radiation, (b) upward shortwave radiation, (c) downward longwave radiation, (d) upward longwave radiation, and (e) net radiation, observed at the 140-m level of the 325-m tower during 1–4 December 2016.**






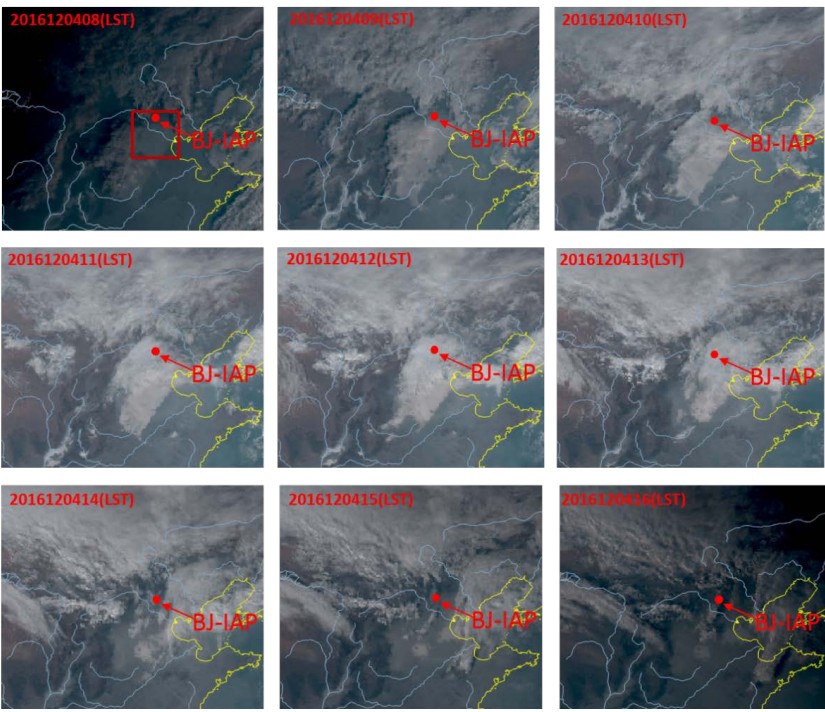


**Figure 8: Hourly Himawari-8 geostationary meteorological satellite cloud images from 0800 LST to 1600 LST 4 December, where the red point marked the location of IAP station in Beijing, and the red square marked the mass of grey.**

In this study, we wanted to focus on the SEB at one level rather than the vertical
difference at between different levels. Moreover, measurements at the 140-m are
above the roughness sublayer layer and are within the surface layer (Miao et al. 2012),
hence only the observations at the 140-m level were used in studying the radiative
exchange. In Fig. 7, the four components shows the daytime pollution received less
shortwave radiation but more longwave radiation than the daytime clean episode. The
*DSR* reduces with gradually worsening air quality on a day-to-day basis. The *DSR*
during this 4-day period reached a peak value (482 W m$^{-2}$) at 1200 LST 1 December.
The differences between the daytime clean and pollution episodes reached about 20
and 110 W m$^{-2}$ at 1200 LST on 2 and 3 December. On 4 December, the largest
difference was 376 W m$^{-2}$ at 1200 LST, followed by 1400 LST (127 W m$^{-2}$), which
approximates that at 1400 LST on 3 December (105 W m$^{-2}$). Overall, compared with
the *DSR* during the daytime clean episode on 1 December, the attenuation ratio of the




*DSR* was about 4%, 23% and 78% at 1200 LST 3–4 December, respectively. Many
efforts have been made on the radiative forcing due to the increasing aerosols loading
by using model simulations and field experiments (Ramanathan et al., 2001; Xia et al.
2007; Ding et al., 2016). Based on observations at the 140-m level at 325-m tower
under eight cloudless days (three clean days and five pollution days) in January 2015,
Wang et al. (2016) found that the maximum attenuation of the *DSR* was 33.7 W m$^{-2}$
and the attenuation ratio was 7.4% at 1200 LST. Due to the difference in solar angle,
degree of pollution, pollutant component, cloud etc., attenuation differences are
expected in different case studies. Here, the *USR* on clean days was larger than in
pollution days with a larger maximum difference (32 W m$^{-2}$) on 4 December, which
was mainly caused by the lower quantity of *DSR* received on 4 December. For the
*DLR*, the diurnal change in the difference between 1 December and 2 December was
insignificant. During the other two daytimes, the *DLR* increased with the
enhancement of pollution level, and the peak values on 3 December and 4 December
were respectively 51 W m$^{-2}$ and 56 W m$^{-2}$.

The diurnal variation of the *DSR* on December 4 was discontinuous, which

suggests the large attenuation of the *DSR* on this day was not only the impact of the
higher aerosol concentrations, but also that of the cloud cover. The largest *DLR* on 4
December also indicated the possibility existence of clouds. Information on the
coverage of clouds can be seen from satellite cloud images, which in this case were
provided by the products of the Himawari-8 geostationary meteorological satellite,
launched by the Japan Meteorological Agency (http://www.eorc.jaxa.jp/ptree/).
According to these data, the first three days were free from clouds (figures are
omitted). From the mass of grey marked by the red square in Fig. 8, it is apparent that
pollutants dominated the BTH region at 0800 LST, and then this area became partially
cloudy. The area over Beijing was covered with cloud at 1000 LST, which lasted
about 3 hours, and then at 1500 LST had become cloudless. Van de Heever and
Cotton (2007) found giant nuclei could lead to strong early enhancement of cloud
development. Moreover, previous studies have found that cloud fraction changes with
aerosol loading (Gunthe et al., 2011; Che et al., 2016). In our case, before the cloudy





day, heavy pollutants occurred over the BTH region, and the IAP station recorded
high relative humidity (> 90%, shown in Fig.4) at midnight, which would have
enhanced aerosol hygroscopic growth, implying significant aerosol–cloud interactions,
referred to Che et al. (2016). Thus, we can deduce that the cloud cover over the BTH
region may in part account for the aerosols on the pollution days, which supports the
abundant cloud condensation nuclei (CNN) for the cloud formation on the following
day. Certainly, further studies with more measurements data and model simulations
are needed to validate this conclusion.

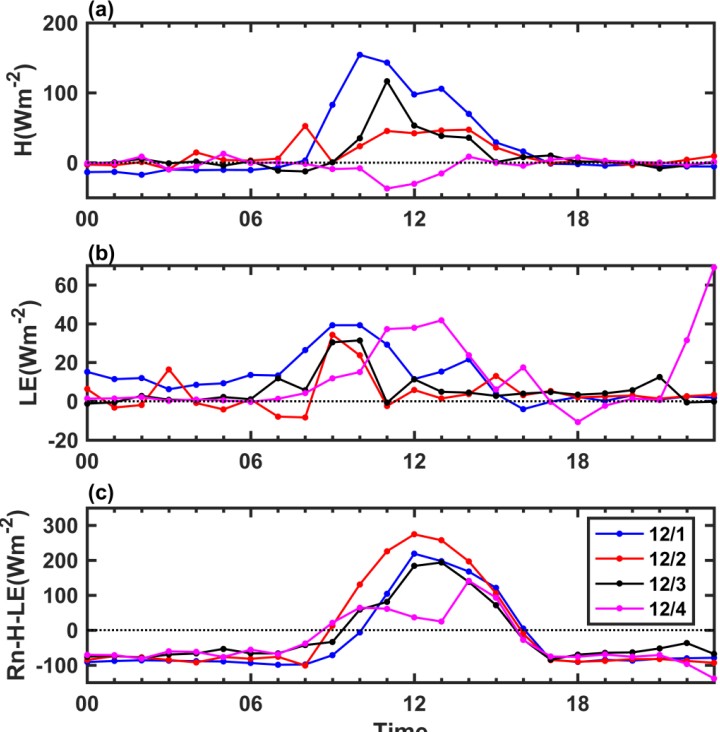

**Figure 9: Diurnal cycle of (a) sensible heat flux, (b) latent heat flux, and (c) heat storage**
**(termed as $R_n - H - LE$), observed at the 140-m level of the 325-m tower during 1–4**
**December 2016.**
In general, the $R_n$ (shown in Fig. 7) attenuation ratio was 2%, 24%, and 86%,
respectively, at 1200 LST 2–4 December. This attenuation of the radiation in pollution
days directly resulted in the change of the SEB, and further induced the change in



structure of the UBL. In Fig. 9, clearly, *LE* was extremely low, at less than 50 W m$^{-2}$
during this 4-day period in winter over the large impervious urban surfaces of Beijing.
The fraction of impervious surfaces around the tower was investigated using an
analytical footprint model and found to exceed 65% (Wang et al., 2015). Mostly,
during daytime, the heat storage was the largest consuming term in the SEB,
accounting for about 65 %, 83 %, 78 % and 71 % averaged in the afternoon hours
(1200–1400 LST) on 1–4 December, respectively. During the early morning on 2
December, the air temperature near the surface (illustrated in Fig. 4) was lower than
on other mornings (i.e., at around 0400 LST, about 5°C lower than on 1 December at
2-m level ABG) and dropped to around zero, meaning a large amount of heat was lost
from the urban volume. Then after sunrise, due to the high thermal conductivity of the
concrete (about 65 times as large as the air), a considerable part of the $R_n$ (maximum
reaching 85% at 1200 LST) was balanced by the heat storage in the urban fabric.
Compared with 1 December, the larger heat storage with similar $R_n$ (differing by less
than 16 W m$^{-2}$) on 2 December led to weaker heat flux, which was negative to the
evolution of UBL and the diffusion of the pollutants, with a slight increasing tend
from 0900 LST to noon, (illustrated in Fig. 2). Specifically, under the conditions of
early morning, much more solar heat is absorbed to warm the large urban fabric after
sunrise, besides, momentum transfer is weak, which leads to less sensible heat. Thus,
in this case, the greater role played by the heat storage in the SEB on pollution days
was responsible for the weakened evolution of the convective boundary layer (CBL).
Compared with the rural surface, Kotthaus and Grimmond (2014) reported the heat
storage in urban surfaces led to delayed warming/cooling, which resulted in the
nocturnal stable conditions generally developing later (Barlow et al., 2015). In our
study, over the urban surface, depending on the differing colder (warmer) early
mornings, heat storage therefore caused a weaker (stronger) development of the CBL.

425   The peak value of the *H* was about 154 W m$^{-2}$, 53 W m$^{-2}$, and 117 W m$^{-2}$, on 1–

3 December, respectively. On 3 December, the heat flux showed a dramatically
decrease, e.g. from 117 W m$^{-2}$ to 53 W m$^{-2}$ in one hour (1100−1200 LST), which
aggravated the negative effect on pollutants diffusion (corresponding to the CS).


There was a thick temperature inversion near to the surface that lasted till the
afternoon on 4 December, as described in last section, which resulted in the
downward heat transfer ($H<0$) to the urban surface in daytime. Gao et al. (2015) also
found that large positive radiative forcing reduced the $H$ and $LE$ by 5–16 W m$^{-2}$ and
1–5 W m$^{-2}$ during a severe fog–haze event over the NCP, by using WRF-Chem
model simulations. By analyzing the measurements collected at a rural site (farmland)
Gucheng in Hebei Province from 1 December 2016 to 31 January 2017 in winter, Liu
et al. (2018) confirmed that the mean daily maximum $H$ was only 40 W m$^{-2}$ on
heavily polluted days (daily mean PM$_{2.5}$ concentration > 150 μg m$^{-3}$), but reached 90
W m$^{-2}$ on clean days (daily mean PM$_{2.5}$ concentration < 75 μg m$^{-3}$). In our case, in
addition to the influence of heat storage in the urban canopy, the large reductions of $H$
on 2–4 December also imply that the high PM$_{2.5}$ (PM$_1$) concentrations from the
nighttime till after sunrise may have caused a fatal influence on the evolution of the
UBL. To improve our understanding of the role of the SEB in air pollution process,
more work is needed, such as consideration of anthropogenic heat flux and the
uncertainty in eddy-covariance observations over complex heterogeneous urban
surfaces. Model simulations have pointed out that the reduced sensible heat resulting
from aerosol backscattering could lower the air temperature and suppress the growth
of the ABL (Yu et al. 2002). Therefore, further and more detailed investigation into
the development of the UBL was reported in the next section.

## 3.4  Development of the UBL

The diurnal cycle of the ABL exerts strong control on the scalar concentrations
of air pollutants (Oke et al., 2017). It is known that the ABL starts to grow after
sunrise, and deepens to a maximum value in mid-afternoon; the whole layer is
convectively unstable and well mixed during daytime, in which layer is defined
as CBL, and this part of the atmosphere is directly affected by solar heating of the
surface. After sunset, accompanied by diminishing turbulence, the boundary-layer
depth declines rapidly, and then the boundary layer becomes to the NBL. Based on the
general changes in BLH, the TKE at a certain depth or the amount of solar radiation,





previous studies have proven that vertical mixing affects pollutants diffusion (Guinot
et al., 2006; Sun et al., 2013; Guo et al., 2017). However, few have documented the
diurnal circle of the intensity variation of vertical mixing in the UBL, on account of
the limitation of instruments. Here, we took advantages of the Dopper lidar (superior
spatial and temporal resolution), to quantify the values of the vertical mixing,
described as vertical velocity variance $\sigma_w^2$ on clean and polluted days.

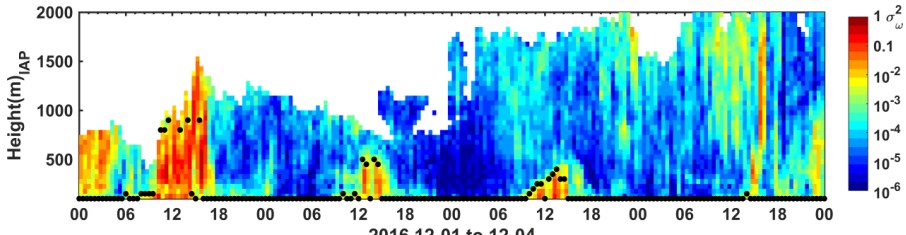


**Figure 10: Velocity variance, $\sigma_w^2$ (m$^2$ s$^{-2}$), calculated from the Doppler wind lidar data.**
**Derived planetary boundary layer depths, based on the threshold method, are depicted as**
**black dots.**
As presented in Fig. 10, it was found that the variance of $\sigma_w^2$ could characterize
the development of the UBL. $\sigma_w^2$ became greater after sunrise (0720 LST), then
reached a maximum at about 1400 LST, exhibiting an obvious trend of decline
( from $\sigma_w^2 > 10^{-1}$ to $\sigma_w^2 < 10^{-2}$ m s$^{-1}$) after sunset (1650 LST). When the UBL developed
into NBL, $\sigma_w^2$ was about $10^{-3}$ m s$^{-1}$ at the 200−300 m levels till midnight and
decreased to about $10^{-4}$ m s$^{-1}$ after midnight until sunset. $\sigma_w^2$ was obviously lower
and its vertical distribution shallower during daytime pollution episodes compared
with the daytime clean episode. On 4 December, the vertical mixing was very weak,
ranging from $10^{-4}$ to $10^{-5}$, and there was barely any diurnal variation of $\sigma_w^2$ till 1500
LST when the PM$_{2.5}$ (PM$_1$) had completely dissipated, which suggests the
radiative cooling of pollutants was a major factor of influence in the UBL
development by suppressing vertical mixing. In return, the stagnating UBL seemed to
act an umbrella, blocking the entrainment with cold-clean air at the upper level, and
solar radiation to the surface, and then further suppressing the diffusion of pollutants,
leading to the increasing PM$_{2.5}$ (PM$_1$) concentration during the CS and much slower
diffusion of PM$_1$ at the surface than that at the 260-m level (Fig. 2). Accordingly, in





our case study, this two-way feedback mechanism between air pollutants and the UBL
is strikingly responsible for the cumulative and dissipation stage of these pollution
episodes in our case.

Compared to 1 December, the vertical mixing is weaker till about 5 hours after

the sunrise on 2 December (CS). This weak evolution of the CBL was consistent with
the weak sensible heat flux (Fig. 9). As discussed in Section 3.3, a large amount of the
heat was trapped in the cold urban fabrics, resulting in poor sensible heat flux after
sunrise on 2 December.

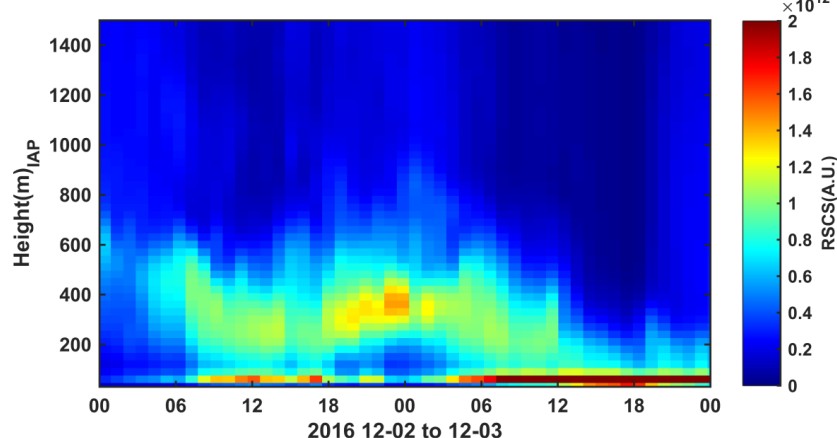


**Figure 11: Evolution of the lidar range-squared-corrected signal (RSCS) at 532 nm from**
**1200 LST 2 December to 1200 LST 3 December 2016. The color scale indicates the intensity**
**of the RSCS, and warm colors represent stronger light scattering.**

Additionally, the $\sigma_w^2$ was mainly ranged from$10^{-6}$ to $10^{-5}$ m s$^{-1}$ ATG to the

detectable observing height during the nighttime from 2200 LST 2 December till the
early morning 0500 LST before the CS on 3 December. This ultra-weak turbulence
transport maintained a very shallow and stable NBL. Note that values of the PM$_1$
concentration (Fig.2) at the 260-m height of the 325-m tower
changed slightly with the time during the ultra-weak turbulence transport periods.
Moreover, before the CS on 3 December, the aerosol lidar data (Fig. 11) showed that
the gradient of the range-squared-corrected signals (RSCS) between the levels of
200–250 m and 400–500 m ATG were larger than the other levels from 1800 LST
(after sunset) 2 December to 0500 LST (before sunrise) 3 December. As we know,





both aerosols and water vapor affect the signals of the lidar. The larger RSCS at the
time mentioned above, in our case, must not only have been because of the water
vapor but also aerosol concentrations, being consistent with the larger $PM_1$
concentration at the 260-m level (more than 100 μg m$^{-3}$). Similarly, the larger RSCS
between the levels of 200–250 m and 400–500 m ATG illustrated these levels were
accumulated with high levels of pollutants and the vertical distribution of pollutants
was inhomogeneous, all of which implies that the 260-m level may have been in the
residual layer. The pollutants in the residual layer are known to play an important role
in the diurnal changes of pollutants at the surface (Hastie et al., 1993; Berkowtiz et al.,
2000; Salmond and Mckendary, 2006). Sun et al. (2013) suggested that the high
concentration of particles in the residual layer could reach the ground the following
morning through convection, causing severe pollutant concentrations in Beijing. In
the Tianjin area, Han et al. (2018) also found that a pollution layer was present at the
altitude of 1000 m in the early morning on 16 December, 2016, where the aerosols in
the higher layers were transmitted to the ground by downward flow before the
formation of heavy pollution. Actually, many studies have focused on this mechanism
of pollutant vertical mixing in a stable NBL from the micrometeorology perspective.
Turbulence in a very stable NBL is typically intermittent and generated by mechanical
shear associated with changes in wind velocity with height (Mahrt et al., 1998),
referred to as upside-down turbulence in an upside-down boundary structure,
compared to the convective daytime case (Mahrt, 1999; Mahrt and Vickers, 2002).
This upside-down structure is characterized by TKE (or $\sigma_w^2$) and turbulent fluxes
increasing with height, and negative transportation of TKE or velocity variances
(Banta et al., 2006). As shown in Fig. 10, the $\sigma_w^2$ became larger at lower levels from
0500 LST 3 December, and then the largest values of $\sigma_w^2$ existed at the 500–600 m,
along with the corresponding $|\Delta U|$ shown in Fig. 6b. This turbulence could transport
the pollutants accumulated in the residual layer downward to the lower levels, and
contributed to the later CS of the pollution. Halios and Barlow (2018) also suggested
that shear production dominates in the upper half of the UBL, and could therefore not
be neglected, even in cases with low wind. Consequently, the intermittent turbulence





generated by the wind shear above a sable UBL plays an important role in the vertical
spreading of pollutants.
As a key variable describing the structure of the UBL, the urban BLH estimated
using the threshold method ($\sigma_w^2 > 0.1$ m$^2$ s$^{-2}$) from the Doppler lidar data is also
shown in Fig. 10. For the CBL, the diurnal variations of CBL height were not
described well by the threshold method for these 4-day, and especially on 4 December
for the weak turbulence on polluted day. Eventually, this empirical method was
derived using data in autumn or summer, during which the vertical turbulence is much
greater than in the winter. In our study, the criterion $\sigma_w^2 > 0.1$ m$^2$ s$^{-2}$ was not
applicable because of weak vertical turbulence transport ($\sigma_w^2 < 0.1$ m$^2$ s$^{-2}$) at certain
times of the day. The threshold method was also invalid in the NBL during this study
period. This may be because of the weak vertical turbulence or smaller height of the
NBL falling below the observable height (100 m). Using Windcube100 data during
summer in Beijing, Huang et al. (2017) also pointed that this method was reasonable
for estimating the CBL depth, while it failed to determine the planetary boundary
layer depths for late-night. Subsequently, they defined the NBL top as the height at
which the vertical velocity variance decreases to 10 % of its near-surface maximum
minus a background variance. However, this new method for the depth of the NBL
also failed in our studied period (figure omitted). This is because the NBL in winter is
mostly steady, which does not satisfy the near-neutral assumption for the method
developed by Huang et al. (2017). Additionally, the NBL has been a major problem
for meteorologists for a long time, especially over polluted urban canopies, which
make the problem far more complex. Therefore, further investigation of this method
should be made in future.
Miao et al. (2018) pointed out that the BLH of a fully developed CBL was
clearly anti-correlated with the daily PM$_{2.5}$ concentration, implying that the change in
the BLH in the afternoon plays an important role in pollution levels, which is similar
with our present. Furthermore, the mixing heights of the fully coupled CBL for 1–,
December were about 900 m, 500 m, and 400 m, respectively. Due to the weaker
mixing intensity on 4 December, it is difficult to capture specific values of the BLH.



As shown in Fig. 2, the maximum daily $PM_{2.5}$ ($PM_1$) concentrations increased
day-by-day from 1 to 3 December, indicating high pollutants concentration near the
surface coincide with a shallow CBL. Petäjä et al. (2016) reported that aerosol–
boundary layer feedback remained moderate at fine PM concentrations lower than
200 µg m$^{-3}$ in Nanjing area, but became intensive at higher PM loadings, and the
BLH reduced to half of the original height at particle mass concentrations slightly
above 200 µg m$^{-3}$. Similarly, particularly strong interactions were verified in the
Beijing area when the $PM_{2.5}$ mass concentration was larger than 150–200 ug m$^{-3}$
(Luan et al., 2018). In our investigation, the low $PM_{2.5}$ ($PM_1$) concentration 46 (48) µg
m$^{-3}$ with a 5% attenuation of $Rn$ reduced the BLH by about 44% on 2 December.
Additionally, for the $PM_{2.5}$ ($PM_1$) concentration of 180 (150) µg m$^{-3}$ on 3 December, a
56% reduction was found with a 25% attenuation of $R_n$. In addition to the $R_n$ term, it
is important to note that the heat storage term in the SEB also makes a significant
contribution to reducing the BLH (details discussed in Section 3.3).

# 580    4   Conclusion

Using data from the 325-m meteorological tower in Beijing and two nearby
lidars, we investigated the characteristics of UBL structure during 1–4 December,
2016 in Beijing and examined the interaction between the structure of the UBL and
the air pollution during three pollution episodes, especially the rapid CS during which
the $PM_{2.5}$ concentration rose from about 100 µg m$^{-3}$ to 500 µg m$^{-3}$ in 12 hours. The
main conclusions can be summarized as follows.
1) During this 4-day study period, the air pollution gradually worsened on a
day-by-day basis, with deceasing surface air pressure. Specially, the large-scale
circulation with a saddled pressure field was highly unfavorable for the dispersion of
pollutants on 3 December during the CS. The RH was larger than 40% during the
pollution episodes, and the vertical distribution of RH showed a remarkably
inhomogeneous pattern during the peak period of the CS with the deep RH (> 80%) at
the 47–240-m levels and heavy surface $PM_{2.5}/PM_1$ concentration (about 500 µg
m$^{-3}$/400 µg m$^{-3}$) in the early morning on 4 December. Temperature inversion ($\Delta\theta > 0$)



occurred during all three nights. For the first pollution episode during the nighttime on
1-2 December, a southern neutral LLJ was found at the 200–1000-m levels after
sunset till midnight over Beijing, which transported the pollutants from the south of
Beijing by advection. For the second episode during nighttime on 2–3 December,
weak southerly wind ($<3$ m s$^{-1}$) dominated below 600-m level, with small vertical
gradients. Meanwhile, for CS on 3 December, there was a very deep and weak wind
layer, which extended to about 1100-m level till 2200 LST 3 December, when the
accumulated PM$_{2.5}$ concentration was larger than 400 μg m$^{-3}$ at the surface.
2) Compared with the *DSR* during the daytime clean episode on 1 December, the
attenuation ratio of the DSR was about 4%, 23% and 78%, respectively, at 1200 LST
2–4 December, which mainly caused a 2%, 24% and 86% reduction of the $R_n$. The
large attenuation of solar radiation on 4 December resulted from the cloud caused by
the large aerosol loading with high RH on 3 December, possibly supporting plentiful
CNN for the formation of cloud. Generally, the latent heat exchange term was very
low during these four days over the urban canopy in Beijing, and the dominate term
was mostly the heat storage, calculated as $R_n - H - LE$ , during daytime, which
accounted for about 65%, 83%, 78% and 71% of $R_n$ (averaged 1200–1400 LST) on 1–
4 December, respectively. We also found that the lower $H$ appeared on the polluted
days than on the clean days, which partly caused by the large consuming term of the
heat storage in the urban fabric.
3) In the CBL, the diurnal circle of lidar-based $\sigma_w^2$ agreed with the variation of
the diurnal cycle of $H$ estimated by the eddy-covariance method at the 140-m level of
the 325-m tower, showing that vertical mixing was obviously weakened on polluted
days. Compared to the clean day, the evolution of the UBL was delayed by about 5
hours after sunrise (about 0720 LST) on 4 December, because of the long-term ($> 12$
hours) existence of temperature inversion resulting from the effects of both aerosols
and clouds. This stagnating UBL seemed to act like an umbrella, suppressing the
diffusion of PM$_1$ at the surface, which was cleaned at about 1500 LST, while the PM$_1$
at the 260-m level was cleaned by the strong clean northerly wind flow at about 0700
LST. Therefore, this two-way feedback mechanism between air pollutants and the




UBL was strikingly responsible for the cumulative and dissipation stage of this
pollution event in our case. Additionally, the intermittent turbulence generated by the
wind shear above the stable NBL in the early morning on 3 December may have
contributed to the CS through the downward transporting pollutants from the residual
layer. Compared to 1 December the reduction of the maximum BLH was 44% on 2
December and 56% on 3 December, whereas, the BLH on 4 December was
unobtainable due to the stagnating UBL growth.

## Acknowledgements

This work was funded by the National Key Research and Development Program of
the Ministry of Science and Technology of China (2016YFC0203304 and
2017YFC0209601).

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
