# Peer review of "Observations of the atmospheric boundary layer structure"

_Atmospheric Chemistry and Physics, 2018_

## Referee Comment (RC1) · Anonymous Referee #3 · 31 Jan 2019

This article provides a complete analysis on the interactions between the air pollutant processes and the structure evolution of urban boundary layer over Beijing during 1-4 December 2016. The synoptic/local meteorological condition, vertical distribution of wind/temperature/humidity, radiative exchange, energy budget, intensity variation of vertical mixing and urban boundary layer depth were investigated by using the data collected from a 325-m meteorology tower and by two lidars. The dataset is unique in a sense that it provides detailed vertical structure of the urban boundary layer during pollution episodes. Besides, one of the nice features of this paper is that it includes analyses of the intensity variation of vertical mixing in a relative superior spatial and temporal resolution. The paper is interesting and generally well-written. However, there are several questions need to be answered before its acceptance.

Major concerns:

1. The boundary layer height calculation with the turbulence method is not applicable for nocturnal condition, as can be seen in figure 10 that most of the BLHs are zeroes at night time. However, nighttime BLHs are specifically important for understanding the PM accumulation. Therefore, correctly retrieval of nighttime BLM is important for the analysis.

2. Temperature inversion is also a key factor to the formation of air pollution. For the case analyzed in the paper, the authors showed that the temperature profiles also had inversions, but how these inversions interact with PM concentration were not analyzed.

3. The maximum heat storage occurred on Dec 2, the BLH reduced significantly from Dec 1 to Dec 2, while the PMs increased only a little, what are their relationships?

Minor questions:

Line 33, 'It clearly showed that weak vertical mixing caused the concentrating of pollutants …', I don't see how this conclusion can be derived, since the concentrating of pollutants is affected by various factors.

Line 57, located on -> located in

Line 83 and 84, delete 'the'

Line 94, 'dormant'? Please reorganized the sentence, it is confusing.

Line 118, 'of the evolution of the vertical UBL structure' -> 'of the vertical UBL structure evolution'

Line 155 - 0n -> on

Line 154 'Aoti surface station' should be defined here.

Line 178, was -> were

Line 190, Acronym 'SEB' should be defined before usage.

Line 202, add 'one of' before 'the most'

Line 248, 'partly led to the' -> 'contributed to the formation of'

Line 257 - Fig.4-> Fig. 4

Line 294 - PM2.5 /PM1-> PM2.5/PM1

Line 327, 'Radiation from the sun' -> 'Solar radiation'

Line 384 - Fig.4-> Fig. 4

Line 393, The LE shows a maximum at 24:00 of Dec 4, why?

Line 441, 'caused a fatal influence on' -> 'suppressed'

Line 451-5, reorganize the sentence.

Line 487, 'is' -> 'was'

Line 500 - Fig.2-> Fig. 2

Line 563 - 1– -> 1

Line 579, 'reducing' -> 'the reduction'

Line 623, 'cleaned' -> 'driven away'

---

## Referee Comment (RC2) · Anonymous Referee #2 · 12 Feb 2019

This paper investigated the dynamics of urban boundary layer (UBL) in Beijing during a severe air pollution episode (1–4 December 2016) and interactions between UBL and surface energy balance (SEB).

Although the topic of this paper fits the scope of ACP, the major drawback lies on the rather isolated analyses: how synoptic conditions, surface energy balance and urban boundary layer evolution interact with each other is not elaborated. Moreover, the feedbacks between these processes are not well summarised, which are expected to appear in the conclusion but unfortunately not.

Also, given the context of this work being a megacity, urban signatures involved in the

urban SEB and PBL dynamics (e.g., anthropogenic emissions, urban morphology, etc.) are largely missing in relevant discussions.

That being said, the separate analyses of synoptic conditions, SEB characteristics and UBL development are sound with detailed descriptions and appropriate comments on relevant studies.

As such, this paper does show potential for publication after the above concerns are well addressed in a revised form.

Specific comments: L253–255: This is not well justified: Traffic emission might not be increased as commuters may be less during weekends than weekdays. L442: It would be good to comment on the possible impacts of anthropogenic heat on the estimates of urban heat storage. L503: the physical meaning of RSCS gradient is better to be provided. Figures 2, 4, 5 and 6: corresponding dates should be explicitly annotated below the x-axes for better legibility.

––––––––––––––––––––––––––

---

## Referee Comment (RC3) · Anonymous Referee #1 · 15 Feb 2019

This is an interesting study about the interactions between pollutant concentration, surface energy budget and PBL evolution over Beijing. I have only a few minor comments that need to be addressed. Detailed points: 1. Line 9. In which sense transportation is affected by air pollution? Usually is the opposite (transportation affects air pollution) 2. Line 186. U, and v are not used in equation (1) and (2). Only w is represented. 3. Line 187. The water vapor density is not in the equations either. Instead, there is $L_v$ 4. Line 190. In which sense the SEB is one dimensional? 5. Line 200. Instead of neglecting the anthropogenic heat flux, I suggest to just analyze the sun of storage and anthropogenic heat flux. 6. Line 216. It is not clear what is the standard deviation between lidars. 7. Figures 8 and 9. The day 4th, during the daytime, the net radiation is

negative (the surface is losing energy through radiation), the sensible heat flux is also negative (the air is hotter than the surface), and the storage term is positive (energy is stored in the surface). How can this be explained? Is there a hot advection to the site?

---

## Author Comment (AC1) · 15 Apr 2019

This article provides a complete analysis on the interactions between the air pollutant processes and the structure evolution of urban boundary layer over Beijing during 1-4 December 2016. The synoptic/local meteorological condition, vertical distribution of wind/temperature/humidity, radiative exchange, energy budget, intensity variation of vertical mixing and urban boundary layer depth were investigated by using the data collected from a 325-m meteorology tower and by two lidars. The dataset is unique in a sense that it provides detailed vertical structure of the urban boundary layer during pollution episodes. Besides, one of the nice features of this paper is that it includes analyses of the intensity variation of vertical mixing in a relative superior spatial and temporal resolution. The paper is interesting and generally well-written. However, there are several questions need to be answered before its acceptance.

Major concerns:

1. The boundary layer height calculation with the turbulence method is not applicable for nocturnal condition, as can be seen in figure 10 that most of the BLHs are zeroes at night time. However, nighttime BLHs are specifically important for understanding the PM accumulation. Therefore, correctly retrieval of nighttime BLM is important for the analysis.

**Responses**: We totally agree with the reviewer that the change of NBL height is very important for the progress of the PM accumulation. However, two methods used in this paper from Doppler lidar data were failed to get the NBL height during wintertime in Beijing, mainly resulted from the limitations of the observable height (100 m) of the Doppler lidar or the suitability of the estimation methods as described in Line 588-603. Therefore, estimating NBL height in polluted steady winter nighttime remains a challenge. Additionally, the height of a fully developed CBL at afternoon, which can be determined reasonably by existing method, also plays an important role in pollution levels (Petäjä et al., 2016; Miao et al., 2018; ). Thus, in our study, we only focus on the change of maximum values of the BLH during noon time day-by-day and its affection on the change of the PM concentration. Further study in the estimation of NBL height may be made with more observational data. Many thanks for this comment.

References:

Petäjä, T., Järvi, L., Kerminen, V. M., Ding, A. J., Sun, J. N., Nie, W., Kujansuu, J., Virkkula, A., Yang, X. Q., Fu, C. B., Zilitinkevich, S., and Kulmala, M.: Enhanced air pollution via aerosol-boundary layer feedback in China, Sci. Rep., 6, 18998, https://doi.org/10.1038/srep18998, 2016.
Miao, Y. C., Guo, J. P., Liu, S. H., Zhao, C., Li, X. L., Zhang, G., Wei, W., Ma, Y. J.: Impacts of synoptic condition and planetary boundary layer structure on the trans-boundary aerosol transport from Beijing-Tianjin-Hebei region to northeast China. Atmos. Environ. 181, 1-11, 10.1016/j.atmosenv.2018.03.005, 2018.

2. Temperature inversion is also a key factor to the formation of air pollution. For the case analyzed in the paper, the authors showed that the temperature profiles also had inversions, but how these inversions interact with PM concentration were not analyzed.

**Responses**: Thanks for this comments. The values of $\Delta\theta$ was used to describe the thermal stability, and the positive values of $\Delta\theta$ during the three polluted nights associated with steady stability which is conductive to the accumulation of the pollutants. Interactions between temperature inversions and PM concentration have been analyzed in previous study. Wang et al. (2019) found a strong positive correlation (R=0.43) between the $PM_{2.5}$ concentration and temperature inversion $(\overline{\theta_{280} + \theta_{320}} - \overline{\theta_8 + \theta_{16}})$. As we know, temperature inversion usually occurs during nighttime with the weak wind in winter time owing to the radiative cooling effect (Li et al., 2018). Moreover, during polluted daytime, when the surface aerosols are accumulated to a certain degree, the aerosols reduce the solar radiation because of the aerosol radiative cooling effect then near-ground temperature subsequently decreases (Fig. 4b), which supports favorable condition with larger difference of temperature (the duration of $\Delta\theta > 0$ increased day by day, Fig. 5a) for the formation of temperature inversion, including radiative cooling effect during nighttime. This interaction between temperature inversion and PM concentration has been discussed in the revised paper (Line 280-287).

References:

Wang, L., Wang, H., Liu, J., Gao, Z., Yang, Y., Zhang, X., Li, Y. and Huang, M.: Impacts of the near-surface urban boundary layer structure on PM2.5 concentrations in Beijing during winter. Sci Total Environ 669: 493-504, 10.5194/acp-2018-1184, 2018.
Li, J., Sun, J. L., Zhou, M. Y., Cheng, Z. G., Li, Q. C., Cao, X. Y., and Zhang, J. J.: Observational analyses of dramatic developments of a severe air pollution event in the Beijing area, Atmos. Chem. Phys., 18, 3919-3935, 10.5194/acp-18-3919-2018, 2018.

3. The maximum heat storage occurred on Dec 2, the BLH reduced significantly from Dec 1 to Dec 2, while the PMs increased only a little, what are their relationships?

**Responses**: We thank the reviewer for pointing out this. BLH has strong influences on the occurrence, maintenance, vertical diffusivity of air pollutants, and the reduced BLH must be a

negative factor, yet not the only one, to the dispersion of pollutants. Here, there two possible reasons can be deduced. One the one hand, the daily $PM_{2.5}$ concentration is not high on 1 Dec, which is limited to be accumulated on 2 Dec, when even present a lower BLH. One the other hand, as mentioned in the introduction part, heavy pollution in Beijing is also highly related to high relative humidity (RH), which is positive to the rapid formation of secondary aerosol. For example, the RH was about 40% after sunrise on December 2, while after sunrise before the CS on December 3, with weak winds, appreciable near-surface moisture accumulation appeared with RH over 60% (Fig. 4 c, a). Based on the previous studies (Tie et al., 2017; Zhong et al., 2019), such enhanced moisture on December 3 would reduce direct radiation through accelerating liquid-phase and heterogeneous reactions to produce more secondary aerosols (Line 268-270) and enhancing aerosol hygroscopic growth to increase aerosol particle size and mass (Kuang et al., 2016), which would back-scatter more solar radiation to space. In contrast, the lower RH on 2 December is thus unfavorable to the formation of secondary aerosol, resulting in the less PM on 2 December concentration. In addition, the sustained stagnant condition on 2 December contributed to a certain degree of the PM concentration before CS, which is one of the preconditions for the rapid formation of CS in later. The discussion has been added to section 3.4 (Line 630-647).

References:

Tie, X., Huang, R.-J., Cao, J., Zhang, Q., Cheng, Y., Su, H., Chang, D., Pöschl, U., Hoffmann, T., Dusek, U., Li, G., Worsnop, D. R., and O'Dowd, C. D.: Severe Pollution in China Amplified by Atmospheric Moisture, Sci. Rep., 7, 15760, https://doi.org/10.1038/s41598-017-15909-1, 2017.
Zhong, J. T., Zhang, X. Y., Wang, Y. Q., Wang, J. Z.i, Shen, X. J., Zhang, H. S., Wang, T. J., Xie, Z. Q., Liu, C., Zhang, H. D., Zhao, T. L., Sun, J. Y., Fan, S. J., Gao, Z. Q., Li, Y. B. and Wang, L. L.. The two-way feedback mechanism between unfavorable meteorological conditions and cumulative aerosol pollution in various haze regions of China, Atmos. Chem. Phys. 19(5): 3287-3306, 10.5194/acp-19-3287-2019, 2019.
Kuang, Y., Zhao, C.S., Tao, J.C., Bian, Y.X., Ma, N., Impact of aerosol hygroscopic growth on the direct aerosol radiative effect in summer on North China Plain, Atmos. Environ., 147(2016), pp. 224-233, 10.1016/j.atmosenv.2016.10.013, 2016.

Minor questions:

Line 33, 'It clearly showed that weak vertical mixing caused the concentrating of

pollutants …', I don't see how this conclusion can be derived, since the concentrating of

pollutants is affected by various factors.

**Responses**: We agree with the reviewer, the original description has been replaced with the

revised sentence 'It clearly showed that vertical mixing was negative to the concentrating of

pollutants'.

Line 57, located on -> located in

**Responses**: Corrected.

Line 83 and 84, delete 'the'

**Responses**: Corrected.

Line 94, 'dormant'? Please reorganized the sentence, it is confusing.

**Responses**: Corrected.

Line 118, 'of the evolution of the vertical UBL structure' -> 'of the vertical UBL structure evolution'

**Responses**: Corrected.

Line 155 - 0n -> on

**Responses**: Corrected.

Line 154 'Aoti surface station' should be defined here.

**Responses**: Corrected.

Line 178, was -> were

**Responses**: Corrected.

Line 190, Acronym 'SEB' should be defined before usage.

**Responses**: The surface energy budget (SEB) has added in Line 195.

Line 202, add 'one of' before 'the most'

**Responses**: Corrected.

Line 248, 'partly led to the' -> 'contributed to the formation of'

**Responses**: Corrected.

Line 257 - Fig.4-> Fig. 4

**Responses**: Corrected.

Line 294 - PM2.5 /PM1-> PM2.5/PM1

**Responses**: Corrected.

Line 327, 'Radiation from the sun' -> 'Solar radiation'

**Responses**: Corrected.

Line 384 - Fig.4-> Fig. 4

**Responses**: Corrected.

Line 393, The LE shows a maximum at 24:00 of December 4, why?

**Responses**: We tried to figure out why a maximum of LE occurred at the midnight on 4 December. The latent flux was estimated by Eq.2 $LE = L_v \overline{w'q'}$, hence we plotted the variations of $w'$ and $q'$ in high frequency from 1300-1500 LST, and 2200-2400 LST on 4 December, and from 2200-2400 LST on 3 December (Fig. R3). We found that $q'$ showed a dramatically increasing from about 2300 to 2320 LST on 4 December, which might be caused by a condensation of water vapour in the atmosphere.

[Figure]

**Fig. R3 Variations of $w'$ and $q'$ in high frequency (10 Hz) from 1300-1500 LST, and 2200-2400 LST on 4 December, and from 2200-2400 LST on 3 December.**

Line 441, ' caused a fatal influence on' -> 'suppressed'

**Responses**: Corrected.

Line 451-5, reorganize the sentence.

**Responses**: These sentences have been rewritten. Thanks.

Line 487, 'is' -> 'was'

**Responses**: Corrected.

Line 500 - Fig.2-> Fig. 2

**Responses**: Corrected.

Line 563 - 1– -> 1

**Responses**: Corrected.

Line 579, 'reducing' -> 'the reduction'

**Responses**: Corrected.

Line 623, 'cleaned' -> 'driven away'

**Responses**: Corrected.

Line 623, 'cleaned' -> 'driven away'

**Responses**: Corrected.

---

## Author Comment (AC2) · 15 Apr 2019

This paper investigated the dynamics of urban boundary layer (UBL) in Beijing during a

severe air pollution episode (1–4 December 2016) and interactions between UBL and

surface energy balance (SEB). Although the topic of this paper fits the scope of ACP, the major

drawback lies on the rather isolated analyses: how synoptic conditions, surface energy balance and

urban boundary layer evolution interact with each other is not elaborated. Moreover, the

feedbacks between these processes are not well summarized, which are expected to

appear in the conclusion but unfortunately not. Also, given the context of this work being a

megacity, urban signatures involved in the urban SEB and PBL dynamics (e.g., anthropogenic

emissions, urban morphology, etc.) are largely missing in relevant discussions.

That being said, the separate analyses of synoptic conditions, SEB characteristics and

UBL development are sound with detailed descriptions and appropriate comments on

relevant studies. As such, this paper does show potential for publication after the above concerns

are well addressed in a revised form.

**Responses**: We thank the reviewer for the overall positive assessment of the paper and for the

constructive and valuable suggestions.

1) In our case, air quality was getting worse with decreasing wind speed. Such weak wind is
   negative to the turbulence transport, which means large heat will be stored in the urban
   canopy, resulting in the weak UBL evolution. Detailed description can be found in Line
   255-259, Line 415-435, Line 463-466, Line 472-474, Line 511-525 and Line 624-647 in the
   revised paper. Moreover, a sketch map, Fig R2 (Fig. 12 in the revised paper) with some
   discussion have been added in section 3.4 in the revised paper to describe the interaction
   between the synoptic conditions (pressure, wind, temperature, relative humidity, etc.),
   surface energy balance and urban boundary layer evolution, and the interactions between the
   aerosol pollutants and UBL structure.

2) We totally agree with the reviewer that the urban signatures involved in the urban SEB and
   PBL dynamics (e.g., anthropogenic emissions, urban morphology, etc.) are significant for
   Beijing (and other megacities). Anthropogenic heat flux ($Q_F$) is an important component of
   the surface energy budget in urban areas. Unfortunately, $Q_F$ is difficult to estimate due to the
   absence of accurate energy consumption and traffic flow data, and they often carry

considerable uncertainty (Sun et al., 2017). Therefore $G - Q_F = R_n - H - LE$ (also suggested by one of anonymous reviewers for this paper) has been analyzed in the revised paper. Besides, urban morphology plays an important role in the SEB and PBL dynamics, however, limited to the detail urban morphology information of Beijing, only the fact that large fraction of the impervious urban surfaces leads to the large urban heat capacity was discussed in the paper (Line 447-451). It should be pointed out that our focus here is on the difference of the heat storage characteristics between clean and polluted days. Was the $Q_F$, an additional energy source, not changed during different days in a short term, the large $(G - Q_F)/R_n$ would imply that more heat is stored in the urban canopy, compared with other term. We found that heat storage ratios during three polluted daytimes was larger mainly as a result of the reduced $Rn$ (caused by the aerosol cooling force), and weaker wind, which contributing to the weaker development of the CBL, compared to the clean daytime. The more detailed discussion has been added in Line 438-451.

Reference:

Sun, T., Kotthaus, S., Li, D., Ward, H.C., Gao, Z., Ni, G.-H., Grimmond, C.S.B.: Attribution and mitigation of heat wave-induced urban heat storage change. Environ. Res. Lett. 12, 114007, 10.1088/1748-9326/aa922a, 2017.

[Figure]

**Fig. R2 (Fig. 12 in the revised paper): Schematic diagrams of the roles of synoptic conditions, surface energy budget in the development of UBL, and the two-way feedback between UBL structure and accumulation of PM$_{2.5}$ during 1–4 December 2016, the values of meteorological elements averaged in noon hours (1200–1400 LST).**

Specific comments:

L253–255: This is not well justified: Traffic emission might not be increased as commuters may be less during weekends than weekdays.

**Responses**: This sentence has been removed.

L442: It would be good to comment on the possible impacts of anthropogenic heat on the estimates of urban heat storage.

**Responses**: We thank the reviewer's valuable suggestion. The more detailed discussion has been added in Line 477-484.

L503: the physical meaning of RSCS gradient is better to be provided.

**Responses**: Thanks for this comment. The meaning of the RSCS has been added in the revised manuscript (Line 543-547).

Figures 2, 4, 5 and 6: corresponding dates should be explicitly annotated below the x-axes for better legibility.

**Responses**: As suggested, all these figures have been revised. Thanks.

---

## Author Comment (AC3) · 15 Apr 2019

This is an interesting study about the interactions between pollutant concentration, surface energy budget and PBL evolution over Beijing. I have only a few minor comments that need to be addressed.

**Responses**: Many thanks for the positive comments and constructive and valuable suggestions.

Detailed points:

1.  Line 9. In which sense transportation is affected by air pollution? Usually is the opposite (transportation affects air pollution)

**Responses**: "The main hazards or negative effects of air pollution generally fall into two categories: human health and transportation". Actually, here the "transportation" is not "wind transportation" but "traffic". To avoid the misleading expression, "transportation" is changed to "traffic".

2.  Line 186. U, and v are not used in equation (1) and (2). Only w is represented.

**Responses**: Corrected.

3.  Line 187. The water vapor density is not in the equations either. Instead, there is Lv

**Responses**: Corrected.

4.  Line 190. In which sense the SEB is one dimensional?

**Responses**: Thanks for good suggestion. The turbulent exchanges of heat are in both vertical and horizontal directions (e.g., Foken, 2008; Leuning et al., 2012). The one-dimensional SEB in the original paper means the idealized formation of the surface energy budget, which ignores the horizontal dimension. Now "The one-dimensional SEB" has been changed to "The SEB without consideration of horizontal advection".

References:

Foken, T.: The energy balance closure problem: an overview, Ecol. Appl., 18, 1351–1367, 2008.
Leuning, R., van Gorsel, E., Massman, W. J., and Isaac, P. R.: Reflections on the surface energy imbalance problem, Agr. Forest Meteorol., 156, 65–74, https://doi.org/10.1016/j.agrformet.2011.12.002, 2012.

5.  Line 200. Instead of neglecting the anthropogenic heat flux, I suggest to just analyze the sun of storage and anthropogenic heat flux.

**Responses**: We agree with the reviewer on this. Now the term $G - Q_F = R_n - H - LE$ is analyzed in the revised paper. Thanks for the comment.

6.  Line 216. It is not clear what is the standard deviation between lidars.

**Responses**: The Doppler lidar can obtain three-dimensional wind, and the vertical velocity variance $\sigma_w^2$ can be used to describe the density of the turbulence. This sentence has changed to "The turbulence method to define the BLH has been proposed by using the Doppler lidar which can obtain three-dimensional wind. The vertical velocity variance $\sigma_w^2$ can be used to describe the density of the turbulence, hence the height of the layer in which vertical velocity variance $\sigma_w^2$ exceeds a given threshold is considered as the BLH".

7. Figures 8 and 9. The day 4th, during the daytime, the net radiation is negative (the surface is losing energy through radiation), the sensible heat flux is also negative (the air is hotter than the surface), and the storage term is positive (energy is stored in the surface). How can this be explained? Is there a hot advection to the site?

**Responses**: The change of net radiation on 4 December illustrated in Fig. 7 was not clear, due to range setting of Y axis. Actually Net radiation was positive during daytime, about 45 W m$^{-2}$ at 1200 LST on 4 December (Fig. R1). Now a line y=0x has been added in the revised Fig. 7e. Sorry for this confusion.

[Figure]

**Fig. R1 Diurnal cycle of net radiation (blue line),sensible heat flux (red line), latent heat flux (black line), and heat storage minus anthropogenic heat flux (termed as $R_n − H –$ LE, purple line), observed at the 140-m level of the 325-m tower on 4 December 2016.**